# A Big Data Platform for International Academic Conferences Based on Microservice Framework

Biao Yang [1], He Liu [1], Xuanrui Xiong [1,*], Shuaiqi Zhu [2], Amr Tolba [3,*] and Xingguo Zhang [4]

1   School of Communication and Information Engineering, Chongqing University of Posts and Telecommunications, Chongqing 400065, China
2   School of Software, Dalian University of Technology, Dalian 116024, China
3   Department of Computer Science, Community College, King Saud University, Riyadh 11437, Saudi Arabia
4   Department of Mechanical Systems Engineering, Tokyo University of Agriculture and Technology, Nakacho Koganei, Tokyo 184-8588, Japan
*   Correspondence: xiongxr@cqupt.edu.cn (X.X.); atolba@ksu.edu.sa (A.T.)

**Abstract:** In the era of the information explosion, big data are always around us. Academic big data are defined as a large amount of data generated in the life cycle of all academic activities, which usually contains a large amount of academic information. Academic conferences can effectively promote academic exchanges among scholars. In recent years, academic conferences in various fields have been held around the world. However, with the increase in the number of academic conferences, the quality of conferences and the efficiency of hosting and participating in conferences are uneven. In today's fast-paced life, high-quality and efficient academic conferences have become the first choice of scholars. In this paper, a conference recommendation method based on a big data analysis of users' interests and preferences is proposed to help users choose high-quality academic conferences and to help organizers reduce conference costs and improve the conference operation efficiency. The method first divides the research fields of user-related academic conferences into three categories: the fields that users are interested in, the fields that users attend, and the research fields that users follow up. Then, the weights of these three categories are set, and the importance of each category recommendation related to the user is calculated. Finally, the conference recommendation index is calculated and several conferences with a high recommendation value are recommended to users. The experimental results show that the proposed conference recommendation method provides a convenient and fast service to conference participants and conference organizers. The developed big data platform can significantly improve the operation and participation efficiency of academic conferences, reduce the costs, and give full play to the role and value of academic conferences.

**Keywords:** academic big data; personalized recommendation; academic conference

## 1. Introduction

Big data analytics [1] is a term created by researchers to describe the use of advanced analytical techniques that process, store, and collect very large diverse big data sets for future examination. Data are being generated at an alarming rate. The rapid progress of the Internet, Internet of Things (IoT), and other technologies [2] is the main reason for this continued growth. The generated data reflect the environment in which the data were generated, so we can use the data obtained from the system to figure out the internal working information of the system. In addition, the growth in the value of data has made big data a high-value target [3].

In the past 20 years, various applications have been developed and the data generated have grown exponentially, which has led to the advent of the era of big data. Big data analytics technology has been applied to many scenarios, such as a topic analysis of online public opinion in social networks [4], traffic guidance [5], and personalized recommendation [6]. Academic big data is a huge data set, including a large amount of academic information,

technical data, and partnership data, that has attracted more and more attention from industry and academia. The widespread adoption of the social computing paradigm has made it easier for researchers to join collaborative research activities and share research results more widely than ever before [7].

### 1.1. Motivation

In recent years, the number of academic conferences held at home and abroad has increased every year, and international conferences have become more and more popular among research scholars. However, with the increase in academic conferences, some of them are not of high quality and cost effectiveness [8]. For example, some conferences had unclear conference topics, unsatisfactory conference documents, and low-level conference papers. Participants rushed to the conference and could not fully interact with other scholars at the conference site. Some amateur conference organizers are unable to organize the conference effectively and waste many conference resources, resulting in a lower cost performance of academic conferences. Therefore, determining how to improve the quality of academic conferences, reduce the cost of organizers, simplify the participation process, and maximize the value of academic conferences has become an urgent problem to be solved.

### 1.2. Research Challenges

There are already some platforms that provide services for academic conferences. For example, there are "AllConferences.Com", "World Conference Calendar", "AAAI Conferences", "EasyChair", "WikiCFP", "Cvent", etc. The conference organizer can quickly publish the solicitation information and participation information through this platform and promote it so that more scholars can see the information faster. Then, attendees can quickly access information about the conference that they want to attend and register for the conference online.

However, most of the existing conference service platforms only provide part of the services mentioned above, rather than providing a "one-stop" service throughout the whole process of academic conferences, and some platforms are neither strong across devices nor international. For example, The WikiCFP platform mainly provides conference solicitation services, but does not support submission and review; the EasyChair platform mainly provides conference solicitation, online submission, and review services, but does not support conference registration and registration services. In this way, the conference organizers release the conference, solicit and review manuscripts, and share conference materials through the cooperation of multiple platforms to complete, and the current practice of the conference organizers is largely to spend the cost of creating a dedicated site to collect information and charges for participants. Moreover, the operations of participants' conference information acquisition, submission, conference registration, and registration also need to be completed on multiple platforms, leading to the cumbersome process of the conference organizer and the conference participant.

### 1.3. Contributions

- A complete set of services related to academic conferences, such as conference release, conference classification and search, conference registration and payment, conference affairs management, etc., is provided, enabling the organizer to realize a "one-stop" for conference hosting and scholar attendance, reducing the cost of academic conference hosting and improving the efficiency of conference hosting and attendance.
- Users can follow the meeting or meeting series that they are interested in. When users follow or register the meeting, the system will remind them of the relevant date of the corresponding meeting, such as the deadline for meeting solicitation, the date of meeting holding, etc. There are two reminding methods—a system message reminder and email reminder—which can make the scholars' time arrangement more reasonable and orderly, effectively reducing the burden of scholars in their busy life.

- A comprehensive evaluation of the conference and its series was conducted by using indicators such as the number of followers, page views, number of participants, rating of participants, and activity of the conference, so as to provide references for scholars to choose participants. The system can recommend personalized conferences to users according to their research interests, attendance records, and meeting attention records so that scholars can quickly find the academic conferences that they are interested in. The system provides optional conference promotion services for the conference organizers, which makes it easier for the conference to attract the users' attention, and promotes the promotion and influence of the conference.
- The system uses a lightweight development framework to achieve a quick response, simple and orderly interface, and smooth transition effect, with excellent user experience. The responsive interface design can adapt to various resolutions of the device, which is convenient for users accessing the system at any time and anywhere.

The rest of this paper is organized as follows: Section 2 reviews related work and Section 3 presents the system design. Section 4 describes the implementation of the system. Section 5 presents the system testing and discussion, and Section 6 concludes the paper.

## 2. Related Work

The development of the social economy and the continuous progress of science have brought more development opportunities in many fields. In particular, the progress of information and communication technology has made it easier for us to access and create data, and has also brought us into a new era of big data. For example, each vehicle generates an average of 30 TB of data in a single day, resulting in a surge in traffic demand from the vehicle to the Internet [9]. Furthermore, driven by the growing demand for real-time mobile application processing, multi-access edge computing is considered to be a promising paradigm for pushing computing resources to the edge of the network [10]. Edge computing is a key technology for solving a series of problems caused by large-scale data transmission [11]. Recent developments in edge computing and content caching in wireless networks enable intelligent transportation systems to provide high-quality services to vehicles [12] while considering the privacy of traffic data [13].

The significant contribution of IoT Industry 4.0 and the growth of IoT-based smart factories have brought new challenges [14] to the effective implementation of big data analytics and machine learning technologies [15]. The IoT is already widely used around the world. Through a large number of diverse devices, such as vehicles, home appliances, smartphones, and environmental sensors connected to the Internet, a large and diverse amount of IoT data can be generated, i.e., IoT big data [16]. Mining the value of the big data of the IoT has good application prospects, such as optimizing urban planning [17], solving the air pollution problem [18], and improving business decision making [19], but it needs an effective analysis system. We have been involved in a heterogeneous data environment, generating and processing various data every day, and the equipment also provides differentiated services [20]. Cloud computing as a large-scale distributed computing paradigm has attracted the attention of the industry and research [21]. In addition, fog computing extends the facilities of cloud computing from the center to the edge network [22]. Despite the advantages of location-aware and low-latency fog computing, the growing demand for ubiquitous connectivity and ultra-low latency is a challenge for real-time traffic management in smart cities [23]. As an integration of fog computing and in-vehicle networks, in-vehicle fog computing is expected to enable a real-time and location-aware network response. By establishing a collaborative service migration framework and analyzing the optimal migration ratio of service collaboration migration [24], vehicular networks can utilize the resources of vehicles and roadside units in order to perform various vehicular applications. Due to the increase in the number of vehicles and the asymmetric distribution of traffic, network operators must design intelligent offloading strategies [25] to improve the network performance and provide high-quality services to users [26].

In recent years, data collected from various sources such as personal devices, the Internet, and the IoT have increased [27]. The recent studies about big data are reviewed in Table 1.

**Table 1.** Studies about big data.

| Ref. | Year | Viewpoint |
|------|------|-----------|
| [28] | 2018 | Allows data owners to outsource their data to remote storage resources provided by various storage providers. |
| [29] | 2019 | A data-driven approach is now a must-have tool for many IoT-based services and applications. |
| [30] | 2020 | Decentralized storage powered by blockchain is becoming a new trend. |
| [31] | 2020 | The existence of wrong data will lead to the deterioration of the quality of big data. |
| [32] | 2021 | Based on research related to privacy issues and the relationship between data innovation through new services and applications in the Internet of Things environment. |
| [33] | 2022 | A data-driven approach is now becoming important to the business, and the Internet of Things data market has emerged. |

With the increase in network scale and the number of users, the application of big data in social networks has emerged [34]. In order to effectively improve the computing power of mobile devices, Wang et al. proposed a method combining mobile edge computing and wireless power transmission technology [35]. With the development of communication technologies and devices, social big data are transmitted among mobile social users through various applications and media, such as through multimedia streaming [36], using healthcare services applications [37], etc. At the same time, the same situation exists in the academic field. The number of academic conferences is increasing, and participants need to choose the content suitable for their research direction [38]. The advent of the information age and the development of network technologies have made it possible to achieve standardized and high-quality online meetings [39]. The informatization of scientific research is increasingly welcomed by researchers, and can also enhance their competitive advantage [40]. Due to the impact of the COVID-19 pandemic, high-quality online conferences have become the preferred choice of many scholars [41]. The ICS platform uses big data analytics to create a new generation of infrastructure and develop new research technologies and applications based on the platform in order to improve the efficiency and innovation of researchers' activities. With big data analytics technology and Internet platforms, organizers and attendees can organize and attend meetings efficiently.

### 3. System

*3.1. System Requirements Analysis*

Requirements analysis is an important stage in the system development process. Sufficient requirements analysis leads to smooth development in the subsequent stages. In this chapter, requirements analyses of the international academic conference service platform are carried out by using UML modeling tools, including a functional and non-functional requirements analysis and feasibility analysis.

The users of the International Academy of Collaborative Professionals include three types of users—non-logged-in user, logged-in user, and system administrator—and they have different system permissions, where the administrator has the highest permissions.

3.1.1. Business Requirements

With the popularity of academic conferences, efficient and high-quality academic conferences are becoming more and more important to scholars. Consequently, there is an urgent need for a fully functional platform to host such conferences.

This study analyzes the development status of the academic conference service platform at home and abroad and summarizes the following core business requirements:

Conference Release and Registration: The conference organizer can publish the conference through the system, including the publication of the call for papers and the publication of the invitation to attend, and can choose value-added services such as conference promo-

tion. Participants can fill in the conference registration information and pay the fee through the system to complete the registration for the corresponding conference.

Conference Management: The person in charge of the meeting can manage the published meetings, including the management of participants, the management of meeting papers, and the issuance of coupons.

Conference Browse and Follow: The system provides users with services such as sorting, filtering, and searching for meetings. Users can easily browse the published meeting information and meeting series in the system and follow the meetings or meeting series of interest to track the corresponding meeting dynamics.

Meeting Date Reminder: The system will alert system messages and emails to users who have followed or registered for the corresponding meeting when the relevant date of the meeting is approaching.

Conference Evaluation: The system can make a comprehensive assessment of the concluded meetings based on several indicators and evaluate the corresponding meeting series.

Conference Recommendations: The system can recommend relevant conferences in related fields to users based on their selected research areas of interest, their conference attendance records, and their conference focus records.

Platform Promotion: The system cooperates with conference organizers. When conference organizers publish the conference through the system, They can obtain discount coupons by introducing the system on the conference website. Users can also obtain discount coupons through providing feedback on their conference experience, proposing improvement measures, inviting new users, and other means.

### 3.1.2. Functional Requirements Analysis

The users of the international academic conference comprehensive service platform include three types: non-logged-in users, logged-in users and system administrators. According to the different permissions, the number of their functions increases.

Non-logged-in users are visitors who have functional requirements that include logging in and registering, retrieving passwords, viewing meeting details, viewing meeting series details, browsing meeting lists, etc.

A logged-in user is one who has a system account and is logged in. In addition to the functional requirements listed above, it also includes personal information management, viewing personal related meeting lists, viewing date reminders, schedule view, message management, publishing meetings, meeting management, meeting concern, meeting registration, meeting promotion, and meeting recommendation.

The administrator is primarily responsible for system management, with functional requirements including system data management, system parameter setting, system information statistics, log management, etc. He/she is also responsible for the maintenance and optimization of the system.

In addition to the above functions, there are some system background functions that are executed regularly, including sending system messages, sending emails, updating meeting recommendation lists, updating meeting ratings, automatic backups, etc.

A dynamic analysis of the system can effectively reflect the internal processes of system functions and the user's operation sequence and results. Activity models are often used as an effective means for a system requirement dynamic analysis, which is usually presented in the form of activity diagrams. An activity diagram shows the sequence of steps of a complex process, such as the algorithm or workflow, which is intended to show the individual steps of the complex process and the order constraints between them. Activity diagrams are object-oriented. Compared with the flow chart, they emphasize the interaction between participants and the system, emphasizing the behavior of the system rather than the processing of the system. The activity diagrams can also show concurrent activity. In this section, we will use activity diagrams to dynamically analyze the interactive functions of the International Academic Conference Service Platform.

The announcement of participation is an essential step for the conference. Its activity diagram is shown in Figure 1. Posting a participation notice is a relatively complex process. Only authenticated users will be able to post participation announcements.

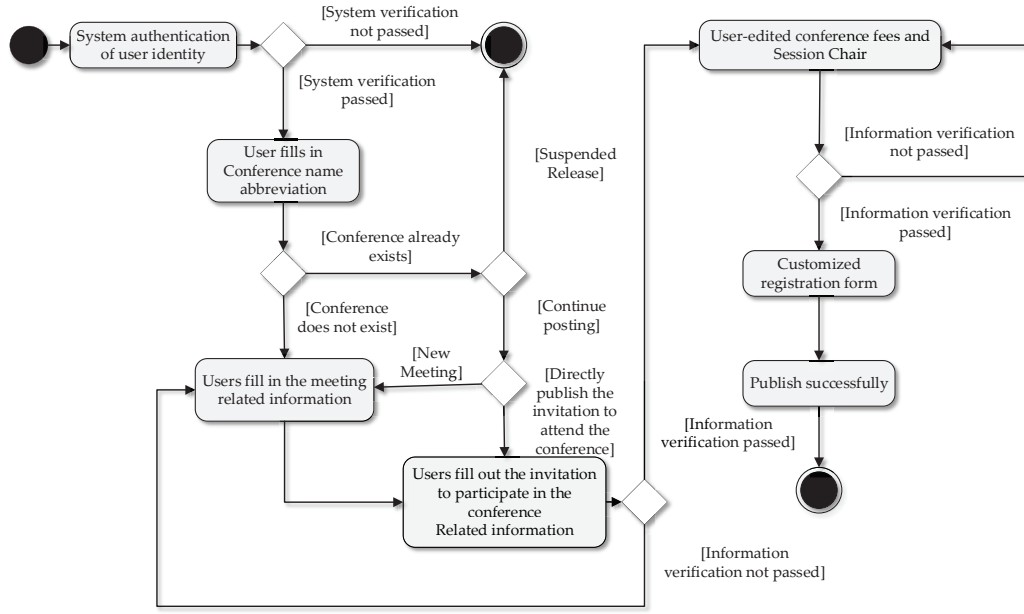

**Figure 1.** Activity diagram of function posting a call for participation.

After the conference organizer publishes the conference notification, users can register for the corresponding conference. The activity diagram of the conference registration function is shown in Figure 2. After entering the conference registration window, users can fill in the registration information, and then the system performs format verification. If it fails, the system will prompt user to modify the failed item. If the authentication passes, the user will select the type of attendees and the optional services provided by the conference. If the selected option does not require payment, the registration will be successful directly. If payment is required, the user pays via PayPal, while the system generates a log of user payments, and then the system prompts the user with the results of conference registration based on the payment results.

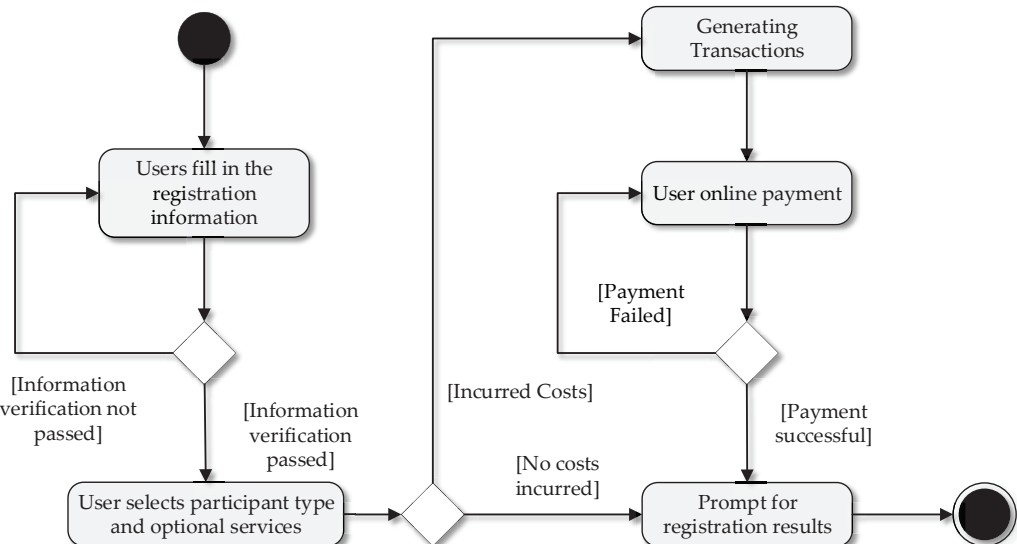

**Figure 2.** Activity diagram of function registering a conference.

The system provides users with a meeting date reminder service. The activity diagram of the function of sending a reminder message regularly is shown in Figure 3.

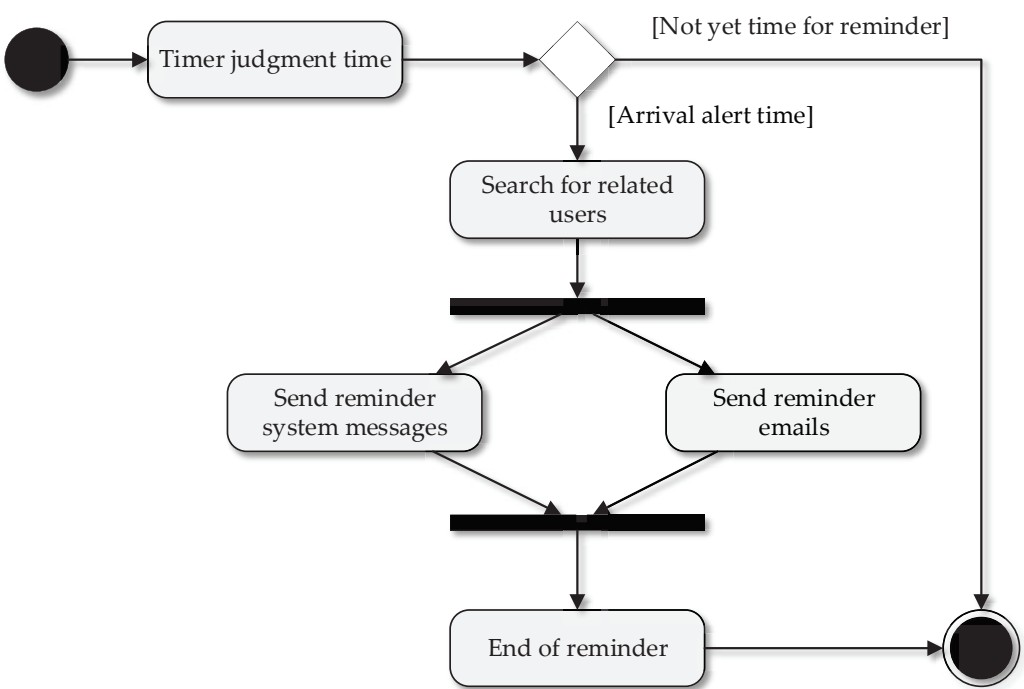

**Figure 3.** Activity diagram of function sending reminding message.

When the reminder time is up, the system timer will send a reminder message to the user, which includes both system reminder messages and an email reminder. As illustrated in Figure 3, when the reminder time is up, the system will send a reminder system message to the user and simultaneously send a reminder email to the user's mailbox. The time interval for reminders is adjusted and set by the administrator.

### 3.2. Functional Module Design

A functional module is a collection of programs that complete a subfunction or business of a system. As a result, they are "low coupled" with each other, meaning that there is a high degree of independence between them. During the development process, changes to one module do not have a large impact on the others, and communication between modules is achieved through the inter-module interface without interfering with the internal implementation of each module.

The functional modularization of a system is to decompose the system function into multiple sub-functional modules, which can be assigned to different developers for development, thus significantly improving the development efficiency. In addition, as there is almost no overlap in the parts that different developers are responsible for, the functional modularization also reduces the difficulty of system integration. Furthermore, independent functional modules will improve the maintainability of the system and provide convenience for future expansion.

As is illustrated in Figure 4, we divided the comprehensive service platform into six functional modules: the user center module, conference release and registration, conference browsing, conference management, timing system module, and administrator module. The division of independent functional modules significantly improves the efficiency of system development, integration, testing, and maintenance.

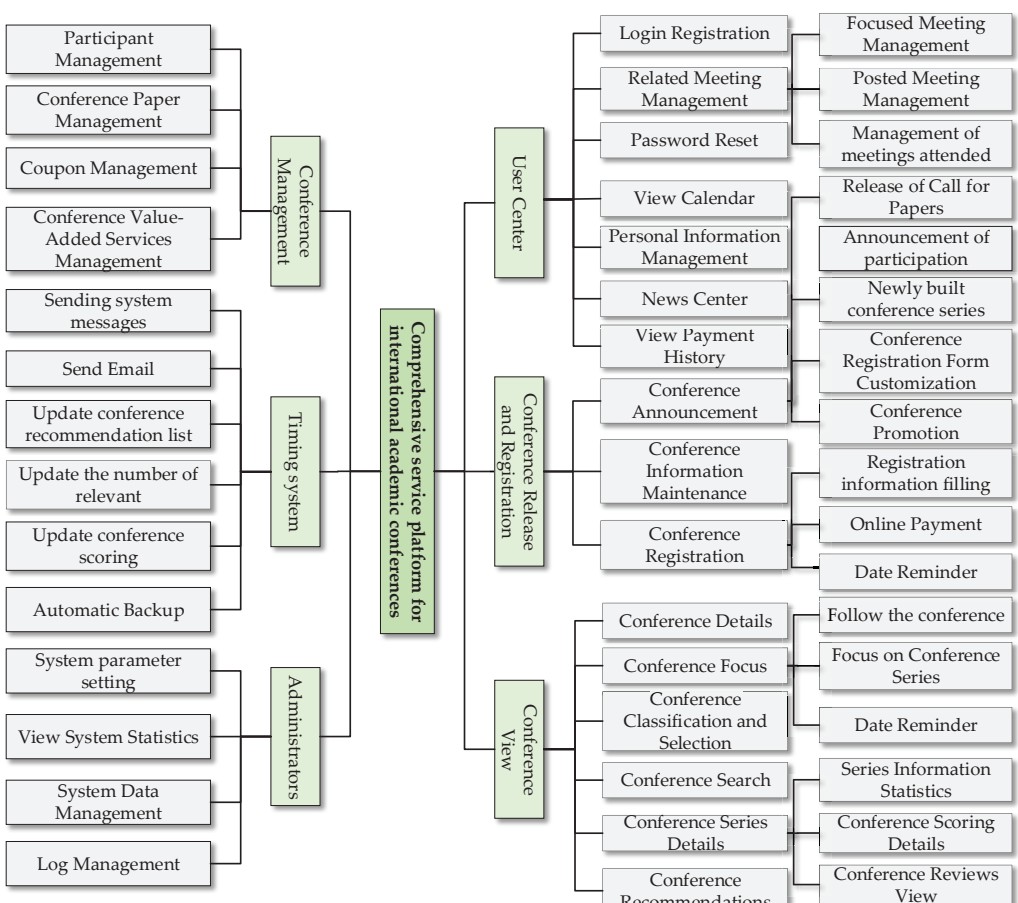

**Figure 4.** Function block diagram of international academic conference service platform.

*3.3. Meeting Personalized Recommendation Design*

In order to facilitate scholars in quickly finding the academic conferences, the system provides users with conference recommendation services [42]. In the system, users can select one to three research fields of their interest in the user's personal center. In addition, when publishing conferences, the system requires the organizer to select one to three research fields for each conference. Therefore, the system can recommend conferences according to the user's interest areas. In addition, the users' participation in conferences and their attention to conferences reflect the tendency of scholars to be interested in academic conferences. Therefore, the user's attendance record and follow-up record can also be used as the basis for meeting recommendations.

In summary, this system's recommendation method considers three categories: the research fields that users are interested in, the research fields of the conferences that they have attended, and the research fields of the conferences that they have followed. We calculated the recommended importance of each category based on the frequency of occurrence. Then, according to the research field of each conference in the system, we calculated the recommendation degree of the conference for the user with the recommendation object and recommend several conferences with a high recommendation degree to the user. The design of this system's meeting recommendation method will be described in detail below.

Firstly, it indicates that the users have established research fields of personal interest. Suppose that the research areas are put together to obtain a set A, which is composed of n research areas $a_i$:

$$A = \{a_1, a_2, .., a_n\} \tag{1}$$

For each domain $a_i$, $f_{i,interested}$, $f_{i,attend}$, and $f_{i,follow}$ denote the number of occurrences of domain $a_i$ in the user's research domain of interest, in the research domain to which the user has attended conferences, and in the research domain to which the user has followed conferences, respectively. The recommended importance $s_i$ of the domain $a_i$ is then calculated according to the following equation:

$$s_i = \alpha f_{i,interested} + \beta \frac{f_{i,attend}}{m_{attend}} + \chi \frac{f_{i,follow}}{m_{follow}} \tag{2}$$

where $\alpha$, $\beta$, and $\chi$ are the weights of the three recommendation categories, and $m_{attend}$ and $m_{follow}$ denote the number of meetings attended and the number of meetings followed by users, respectively. Generally, the research areas of interest set by users in the system's personal center best reflect their research interest preferences and directly reflect their desire to attend academic conferences in those research areas. As time passes, scholars' research interests or research directions may change. At this point, the user can modify their research area of interest set in the system, while the participation record cannot be changed. Therefore, the user's participation record is less informative than their own choice of the research area of interest for a recommendation. Users can follow multiple meetings in the system at the same time, and, in some cases, may not have sufficient knowledge of the meetings that they follow, or these meetings may only be candidates for the user's participation. Therefore, the weight of the user's following record will be smaller than the first two recommendation bases. We designed the relationship of the three weights as $\alpha > \beta > \chi$.

After calculating the recommended importance of the user's relevant research area, the recommendation degree of the conferences in the system that have not yet been held and have not reached the submission deadline was calculated. Suppose that the research area to which a conference belongs is $b_j$, where j takes the values 1, 2, and 3. Then, the recommendation degree $S_j$ for each domain of the conference for the recommended target users is taken as follows:

$$S_j = \begin{cases} 0 & \text{if } b_j \notin A \text{ or } b_j \text{ is None} \\ s_i & \text{if } b_j \in A \text{ and } b_j = A \end{cases} \tag{3}$$

where $b_j$ not existing is a case where the conference publisher has not selected enough research areas for three affiliations at the time of publishing the conference. Then, the conference recommendation degree R for the conference for the corresponding user is calculated as follows:

$$R = \sum_{j=1,2,3} S_j \tag{4}$$

Conferences are sorted from the highest to lowest recommendation. If there are multiple sessions with the same recommendation, they are sorted by their number of followers from most to least. If the number of followers is equal, they are sorted by their number of views from most to least. Finally, the system will recommend the top-ranked sessions to the corresponding users.

The above discussion is the case where all three recommendation categories exist; however, there are users who have not refined their personal research areas of interest, attending conferences through the system. If at least one of the three recommendation categories exists, Equation (1) through (4) still apply. This is because the frequency of the occurrence of the study area included in the non-existent basis is 0, and the corresponding recommended importance calculated is also 0. Therefore, even if one of the recommendation bases exists, set A of research areas associated with the user is not empty. For users for whom none of the three recommendation bases exist, the system recommends conferences

with customized promotion services and conferences with a high number of followers and views to them.

### 3.4. Static Structure Design

The static structure of a system is designed to represent both its internal properties and the connections between them. Class modeling is a common approach used for designing such a structure.

This system has 183 classes for all modules, including 28 entity classes and 33 test classes. The control classes, persistence classes, and entity classes have a one-to-one correspondence. They work together at different levels to achieve the system's functions. Because of the large number of classes, we divided the core classes of the system into two major parts according to their functional connections, and each is represented as a class diagram.

### 3.5. Dynamic Model Design

In a system, no object exists in isolation; if it is isolated, it will lose its meaning of existence. All of these objects interact through messaging; thus, when designing a system, it is inevitable to model the interactions dynamically [43] and present them in the form of timing diagrams. UML timing diagrams were used to model interactions between participants and system objects.

In this section, the core use cases of the system are analyzed with timing diagrams and scenario descriptions. Figure 5 shows the timing diagram of the conference concerns for logged-in users, while Figure 6 shows the timing diagram of the logged-in user posting the participation notice.

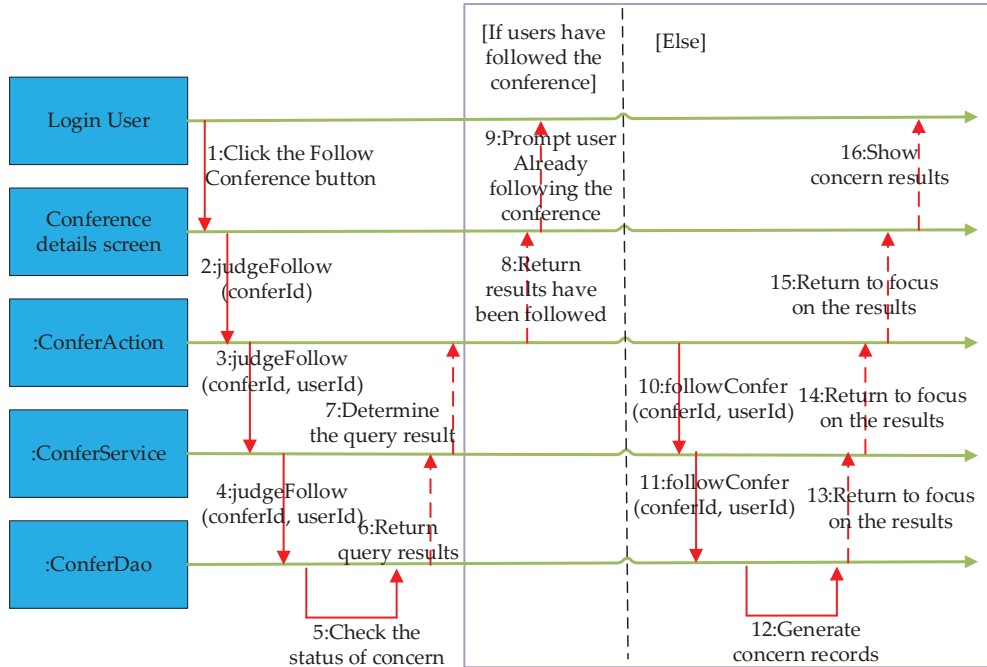

**Figure 5.** Sequence diagram for conference following of logged-in user.

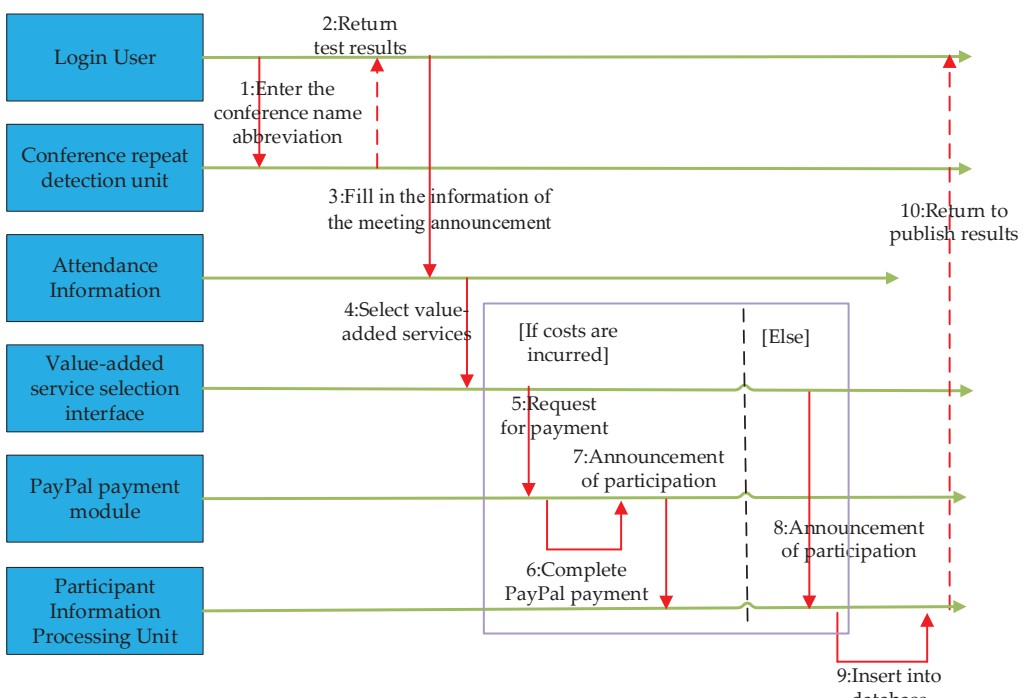

**Figure 6.** Sequence diagram for posting a call for participation of logged-in user.

## 4. System Implementation

After the requirements analysis and system design stage, the system development will enter the system implementation stage. System implementation is an important part of software development that directly determines the final quality of the software. This chapter describes the implementation of some system functions, as well as showing some core code and system interfaces.

### 4.1. Personalized Recommendations for Meetings

The system regularly executes the recommendation algorithm to update the conference recommendation table. Based on the recommendation table, the system can provide users with personalized academic conference recommendations. This function is also executed when the user clicks on the "Meeting Recommendations" item field in the personal center.

The function first calls the getAreaRecommend function to determine if the recommendation field associated with the user is empty. If it is, personalized meeting recommendation will not be provided for this user. If not, the recommended importance of these related fields is calculated. Then, using the getRecommendDegree method, the recommendation degree of each unconference in the system is calculated for the currently traversing user using the conference ID as the key and the recommendation degree as the value, and the results are encapsulated using the data structure map. Finally, the conferences in the map are sorted in descending order of recommendation, and a number of conferences with the highest recommendation are taken to update the recommendation table in the database.

If the research area to which the conference belongs is in the set of user-related recommendation areas, the recommendation degree is accumulated; otherwise, the recommendation of the session to the corresponding domain is set to 0. Finally, the recommendation of the incoming session to the incoming user is returned, and the interface of personalized recommendation results for meetings is shown in Figure 7, with the number of recommendations set to 3.

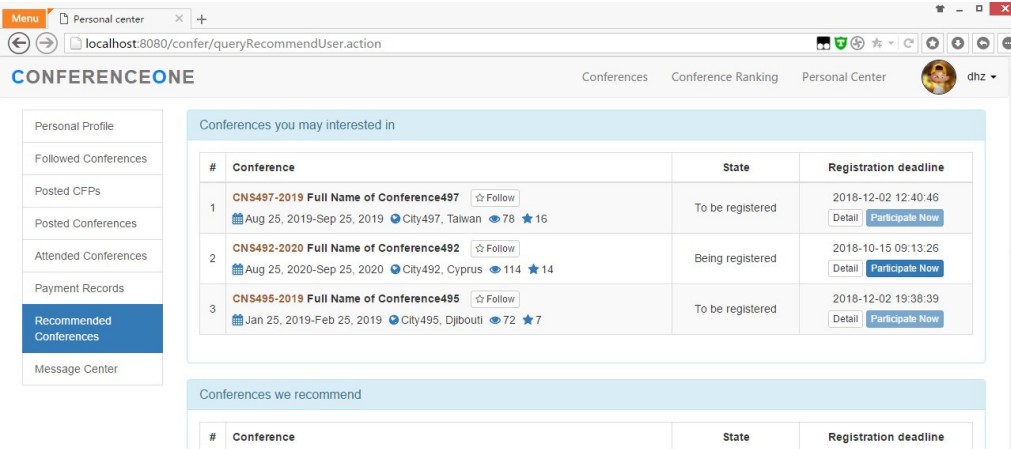

**Figure 7.** UI of conferences personalized recommendation.

### 4.2. Conference Series Information Visualization

In order to enable users to obtain an overview of the conference series information and the previous session information of the conference series, the system provides a visual display of the conference information under the conference series. This section outlines the process of generating the chart of change in conference scores and a chart of the spatio-temporal distribution of conferences.

The change in conference rating under conference series can reflect the trend of the conference quality change and provide a reference for users to choose which conferences to attend.

The front end utilizes ECharts.js for visual graphing. Firstly, the Ajax asynchronous method is used to obtain and parse the data, construct the horizontal and vertical axes of the line graph, the series values, and the hover hint, configure the style properties of the graph, and then draw it. As shown in Figure 8, a graph of the change in meeting scores under a meeting series is presented.

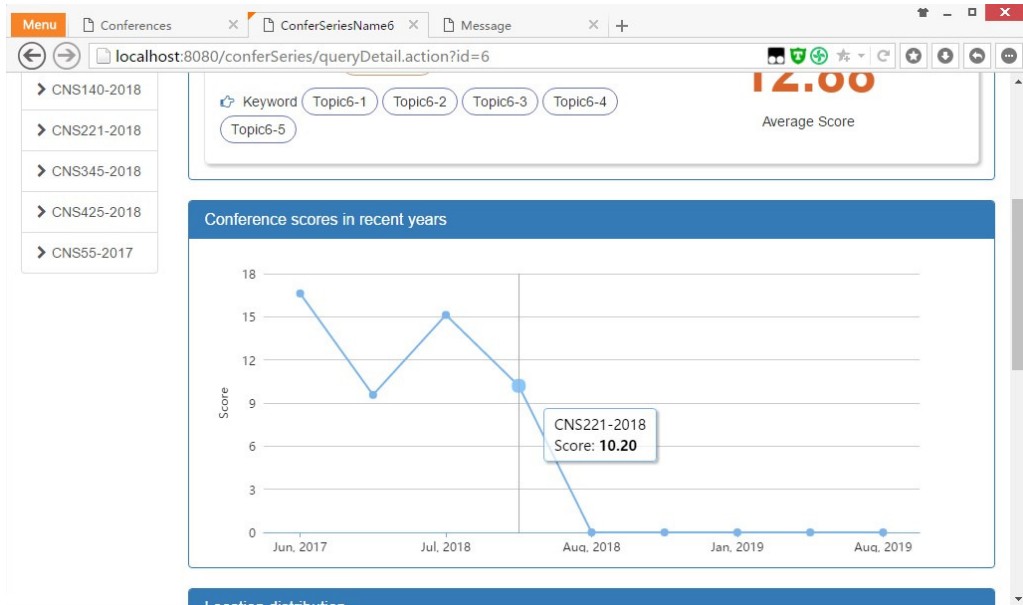

**Figure 8.** Score variation diagram of conferences in a conference series.

The process of generating the spatio-temporal distribution of meetings under a conference series is similar to that of the fraction change graph. The geographical latitude and longitude information of the meeting is also required. The system stores the names of countries, provinces, and cities in the world, as well as their latitude and longitude information in the database. If there is no corresponding city record in the database table, the average latitude and longitude of the country where the meeting is held will be used as the latitude and longitude of the meeting site.

## 5. System Testing

System testing is an essential part of software development and a direct guarantee for the system to run smoothly after it goes online. This chapter will introduce the testing methods, testing process, test cases, and results of the international academic conference comprehensive service platform in the testing phase.

### 5.1. Test Objectives and Methods

To ensure that the system meets the design requirements, the test shall be carried out during and after the coding process.

During the system implementation phase, the Java testing framework Junit was used to perform unit testing on a function-by-function basis. We used printouts and breakpoint debugging to test the structure of loops and branches within functions, calls between functions, and data interactions between layers to ensure that no errors occur in the program. Additionally, the actual running environment was simulated, the parameters corresponding to the test cases were input, and the function was judged to be correct or not by the output results. Junit has an assertion function that compares the actual result with the expected result, and directly determines whether it is correct, thus facilitating the developer to judge the output result positively or negatively and improving the testing efficiency. The method of testing the internal logical structure of a program belongs to white-box testing.

Based on the completion of the unit testing of each functional module, the integration of each module and the front and back of the system was carried out. During the integration process, integration testing of the system is required to ensure the correctness of information interaction between modules and data transfer between the front-end and back-end. The judgment of test results is mainly based on the correspondence between input data and output data, which is a kind of black-box testing. To achieve this, test cases including all possible scenarios are written and each interface element and each system function is tested with visual representations such as interface jumps, system prompts, running logs, background output results, and database record changes.

### 5.2. Testing Process

During the coding process, the developed system functional modules will be tested. Furthermore, once the coding is finished, the entire system will be tested. To ensure that the integrated service platform meets its functional and non-functional requirements, we conducted extensive testing on the platform, mainly consisting of unit testing and functional and non-functional testing.

In unit testing, we first designed test cases for each sub-function of the system, including normal data and abnormal data. Then, we wrote test methods for unit testing. The test cases for functional tests are similar to those of unit tests and should take into account as many cases as possible. Functional testing is mainly performed through interface operations, such as inputting and clicking according to the test cases. Non-functional testing can also be carried out through interface operations or by using testing tools such as Jmeter, which simulates the actual environment of the system's formal online operation to test its non-functional requirements. The unit testing of this system is mainly conducted on the methods of the service layer.

*5.3. Test Cases and Results*

5.3.1. Unit Testing

In the layered structure of this system, the service layer calls the methods in the DAO layer, the DAO layer calls the methods in the map profile, and the map profile uses SQL statements to access the database and then returns the query results layer by layer. For unit testing, Java's Junit testing framework is used to write test methods, construct parameters to call the underlying function, and then trace the test through breakpoints. The printed results or database record changes are observed to determine whether the system function is correct.

This test method is used to test the database operation part of the user data insertion function. By adding @Test to the method header, it can be run directly through Junit without writing the main method. Testers can test functions for exceptions by writing Junit test methods while writing individual business functions, thus laying a good foundation for the integration and testing of subsequent functional modules.

5.3.2. Integration Testing

The business logic of the system has multiple conditional branches. To ensure the functional and logical correctness of each branch, an overall integration test of the system is required. This is mainly carried out by writing test cases and then judging the correctness of the test cases based on the results of operations such as input and click in the interface. As space is limited, this section is illustrated by test cases and results of a few representative functions.

The results of testing the conference sorting and classification function are presented in Table 2. The test procedure focused on testing the multi-conditional combination query of the database and the paging function of the returned results.

**Table 2.** Test cases and results of function conference classifying and sorting.

| Use Case Number | Test Contents | Expected Results | Test Results |
|---|---|---|---|
| A001 | The default classification and sorting modes are used to display meetings. | All meetings are displayed in ascending order by conference start time. | Pass |
| A002 | Different classification conditions and sorting methods are selected. | The meetings under the filtering conditions are displayed in the specified sorting mode. | Pass |
| A003 | Page turning and display when the query result is divided into multiple pages. | The meeting is displayed on multiple pages. No duplicate meeting records are displayed on different pages. | Pass |
| A004 | The number of records and the total number of pages displayed when the query result is less than one page. | The record is displayed in one page, the total number of pages is displayed as 1, and the page cannot be turned. | Pass |
| A005 | There are no meeting records under the search conditions. | No record prompts are displayed, and the paging section and the sort button are also not displayed. | Pass |
| A006 | The page is refreshed to check whether the records corresponding to the page number before refreshing are displayed. | The records corresponding to the page number before refreshing are displayed. | Pass |

The results of testing the conference message refinement function are shown in Table 3. The testing process focused on testing the asynchronous data interaction function of the parent–child interface, as well as the form validation function, which included validating required fields and verifying the legitimacy of user input content.

**Table 3.** Test cases and results of function conference information filling.

| Use Case Number | Test Contents | Expected Results | Test Results |
| --- | --- | --- | --- |
| A007 | If you do not enter any required items or enter several spaces, click the Submit button. | The color of the corresponding input box becomes red and the focus is obtained. | Pass |
| A008 | Required if no value is selected, click the Submit button. | The color of the corresponding selection control box changes to red and the focus is gained. | Pass |
| A009 | Enter the numeric entry string and click the Submit button. | The corresponding number input box turns red and gains focus. | Pass |
| A010 | Enter the illegal email format and click the Submit button. | The corresponding mailbox input box turns red and gains focus. | Pass |
| A011 | Enter content beyond the limit length and click the Submit button. | The input box that exceeds the content limit turns red and gains focus. | Pass |
| A012 | After conference series data are released, the system automatically completes related information on the page. | Conference series name, research field, and keyword information automatically complete. | Pass |

The test cases and results of the conference registration deadline reminder function are presented in Table 4. This test procedure is mainly aimed at verifying the effectiveness of the system triggers and the accuracy of the timed tasks' trigger time.

**Table 4.** Test cases and results of function conference registration deadline reminding.

| Use Case Number | Test Contents | Expected Results | Test Results |
| --- | --- | --- | --- |
| A013 | When the meeting is followed by at least one user, the registration deadline is reached. | A reminder message is sent to users who are interested in the meeting. | Pass |
| A014 | The meeting was attended by at least one user and the registration deadline was not met. | No reminder messages are sent. | Pass |
| A015 | The meeting was not followed by any user and the registration deadline was reached. | No reminder messages are sent. | Pass |
| A016 | The meeting was deleted and the registration deadline was reached. | No reminder messages are sent. | Pass |
| A017 | When the meeting is followed by at least one user, the registration deadline is reached. | The background generates an email sending failure log. | Pass |

## 6. Conclusions

Academic conferences provide a useful platform for scholars to communicate with each other and contribute to academic development. This paper proposes a conference recommendation method based on a big data analysis of user interest and preference. Furthermore, an online academic conference service platform was constructed to assist users in selecting suitable academic conferences and to help conference organizers enhance the efficiency of operating academic conferences and increase conference attendance.

The conference recommendation method is based on big data analysis, using users' historical data, research fields, and interests, and providing convenient personalized conference recommendation services for participants. At the same time, it will also comprehensively evaluate indicators such as the number of people paying attention to the conference, the number of views, the number of participants, the score of the participants, the activity of the conference, and the series of the conference, providing a reference for scholars to choose to participate. The design and implementation of the academic conference comprehensive service platform is based on the needs of conference organizers and participants. The platform provides various services for conference organizers and participants, including conference release, conference classification and search, conference registration and payment, conference management, conference recommendation, conference attention and agenda reminder, conference evaluation, and conference promotion. The experimental

results show that the online academic conference service platform based on the proposed conference recommendation method can provide convenient and fast conference organization and participation services for conference organizers and participants and significantly improve the running and participation efficiency of the conference which can enable scholars to arrange events more reasonably and in an orderly manner, and effectively reduce the burden of scholars in their busy life.

The system has completed the development of the international academic conference comprehensive service big data platform, but there are still works to be improved. In the future, we will further improve the conference recommendation algorithm, incorporate more indicators into the algorithm, adopt more reasonable calculation methods, and provide personalized conference recommendation services for users. Not only will conferences in the scholar's field of study be recommended, but similar fields and future possible fields of study will also be recommended so that scholars can quickly obtain high-quality conference information and broaden their research fields.

**Author Contributions:** Conceptualization, B.Y. and H.L.; methodology, S.Z.; software, X.X.; validation, X.X., A.T. and X.Z.; formal analysis, B.Y.; investigation, H.L.; resources, A.T.; data curation, B.Y.; writing—original draft preparation, B.Y.; writing—review and editing, X.X.; visualization, H.L.; supervision, S.Z.; project administration, X.Z.; funding acquisition, A.T. All authors have read and agreed to the published version of the manuscript.

**Funding:** This work was funded by the Researchers Supporting Project number (RSPD2023R681), King Saud University, Riyadh, Saudi Arabia.

**Data Availability Statement:** Not applicable.

**Conflicts of Interest:** The authors declare no conflict of interest.

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
