# Peer review of "A Big Data Platform for International Academic Conferences Based on Microservice Framework"

_electronics, doi:10.3390/electronics12051182_

Round 1

Reviewer 1 Report

The manuscript looks good and useful topic. It would be great if the authors add some discussion about those multidisciplinary conferences whom authors are not following, not in the exact same field and do not have information for attending.  

Author Response

Response to Reviewer 1's Comments

Comment 1: The manuscript looks good and useful topic. It would be great if the authors add some discussion about those multidisciplinary conferences whom authors are not following, not in the exact same field and do not have information for attending. 

Response 1: Thank you for your suggestion. As you mentioned, we have only recommended conferences in the similar field, and we are now doing precise matching based on keywords. Our future work is to further improve the conference recommendation method, intending to recommend cross-disciplinary conferences and related field conferences, so that scholars can be well matched with interested conferences. We have supplemented the future work in the conclusion, i.e.,

"The system has completed the development of the international academic conference comprehensive service big data platform, but there are still spaces to be improved. In the future, we will further improve the conference recommendation algorithm, incorporate more indicators into the algorithm, and provide personalized conference recommendation services for users. Not only will conferences in the scholar's field of study be recommended, but also similar fields and future possible fields of study can be recommended, so that scholars can precisely obtain high-quality conference information and broaden their research fields. "

Reviewer 2 Report

Let me immediately say this is an interesting research piece and I enjoyed reading it.

I have very few comments to implement for getting it published:

 1.      Your discussion of related work (page 2) may benefit from citing the following recent publications on the topic

Giulio Ferrigno, Nicola Del Sarto, Andrea Piccaluga, Alessandro Baroncelli. 2022. A bibliometric analysis of Industry 4.0 base technology and business models. Boosting knowledge & trust for a sustainable business.

Dagnino, G. B., Picone, P. M., & Ferrigno, G. (2021). Temporary competitive advantage: a state‐of‐the‐art literature review and research directions. International Journal of Management Reviews, 23(1), 85-115.

Ferrigno, G., & Cucino, V. (2021). Innovating and transforming during COVID‐19: insights from Italian firms. R&D Management, 51(4), 325-338.

 2.      I see so many figures. Are all of them necessary? Perhaps you can eliminate few of them.

3.      In the conclusion, theoretical contributions might be reinforced.

I hope you found my comments constructive. Thanks again for giving me the possibility to read your paper. Look foeward to seeing it published 

Author Response

Response to Reviewer 2's Comments

Comment 1: Your discussion of related work (page 2) may benefit from citing the following recent publications on the topic. Giulio Ferrigno, Nicola Del Sarto, Andrea Piccaluga, Alessandro Baroncelli. 2022. A bibliometric analysis of Industry 4.0 base technology and business models. Boosting knowledge & trust for a sustainable business. Dagnino, G. B., Picone, P. M., & Ferrigno, G. (2021). Temporary competitive advantage: a state‐of‐the‐art literature review and research directions. International Journal of Management Reviews, 23(1), 85-115. Ferrigno, G., & Cucino, V. (2021). Innovating and transforming during COVID‐19: insights from Italian firms. R&D Management, 51(4), 325-338.

Response 1: Thank you for your suggestion. We have added references [14], [40], and [41] to the relevant work in the revised manuscript.

Comment 2: I see so many figures. Are all of them necessary? Perhaps you can eliminate few of them.

Response 2: Thank you for your suggestion. We has deleted Figures 5, 6, and 11, which have less information.

Comment 3: In the conclusion, theoretical contributions might be reinforced.

Response 3: Thank you for your suggestion. We have revised the conclusion and added some theoretical contributions. The content we added is:

"The conference recommendation method is based on big data analysis, using users' historical data, research fields, interests, and providing convenient personalized conference recommendation services for participants. At the same time, it also comprehensively evaluates the indicators, such as the number of people paying attention to the conference, the number of views, the number of participants, the score of participants, the activity of the conference, and the series of the conference, providing reference for scholars to choose for participation. The design and implementation of the academic conference comprehensive service platform is based on the needs of conference organizers and participants. The platform provides various services for conference organizers and participants, including conference release, conference classification and search, conference registration and payment, conference management, conference recommendation, conference attention and agenda reminder, conference evaluation and conference promotion. The experimental results show that the online academic conference service platform based on the proposed conference recommendation method can provide convenient and fast conference organization and participation services for conference organizers and participants, and significantly improve the running and participation efficiency of the conference, which can enable scholars to arrange events more reasonably and orderly, and effectively reduce the burden of scholars in their busy life."

Reviewer 3 Report

1) Research title "A Big Data Platform for International Academic Conferences  Based on Microservice Framework" give good idea about conference management system but same times, i feel that comparison with the real time management system is missing. Author must include comparative with his platform with latest existing platforms.

2) Author must used marketing strategies approach in conference management system.

Author Response

Response to Reviewer 3's Comments

Comment 1: Research title "A Big Data Platform for International Academic Conferences  Based on Microservice Framework" give good idea about conference management system but same times, i feel that comparison with the real time management system is missing. Author must include comparative with his platform with latest existing platforms.

Response 1:Thank you for your advice. We have adjusted the content of the introduction to add  research challenges. Research challenges content is as follows:

There are already some platforms that provide services for academic conferences. For example, there are "AllConferences.Com", "World Conference Calendar", "AAAI Conferences", "EasyChair", "WikiCFP", "Cvent", etc. The conference organizer can quickly publish the solicitation information and participation information through this platform, and promote it, so that more scholars can see the information faster; Attendees can quickly access information about the conference they want to attend and register for the conference online.

However, most of the existing conference service platforms only provide part of the services mentioned above, rather than providing a "one-stop" service throughout the whole process of academic conferences, and some platforms are not strong across devices and international. For example, The WikiCFP platform mainly provides conference solicitation services, but does not support submission and review; The EasyChair platform mainly provides conference solicitation, online submission and review services, but does not support conference registration and registration services. In this way, the conference organizers release the conference, solicit and review manuscripts, and share conference materials through the cooperation of multiple platforms to complete, and most of the current practice of the conference organizers is to spend the cost of creating a dedicated site to collect information and charges for participants. Moreover, the operations of participants' conference information acquisition, submission, conference registration and registration also need to be completed on multiple platforms, leading to the cumbersome process of the conference organizer and the conference participant.

Comment 2: Author must used marketing strategies approach in conference management system.

Response 2: Thank you for your suggestion. In Section 3.1.1, we have added some promotional content for the platform, i.e.,

"Platform Promotion: The system cooperates with conference organizers. When conference organizers publish the conference through the system, They can obtain discount coupons by introducing the system on the conference website. Users can also obtain discount coupons through providing feedback on their conference experience, proposing improvement measures, inviting new users, and other means.”

Reviewer 4 Report

In this paper A Big Data Platform for International Academic Conferences Based on Microservice Framework has been proposed .Overall, I found that this is a good work and novelty of the paper is also found satisfactory, however I would suggest following changes in it

1. In abstract " The experimental results show that the proposed conference recommendation method provides convenient and fast participation services as well as conference organization services. The developed big data platform can significantly improve the operation and participation efficiency of academic conferences, reduce the costs, and give full play to the role and value of academic conferences"  there is no quantataive data here to prove that our model is better is how much percentage and how much does this can reduce cost ??

2. In section 1, Different types of algorithms and techniques used for same issye is missing , add some more details in introduction part’?

3.  no contribution points can be seen in the mansucirpt ?.

4. Section 2 presents some of the important literature work related to , it would be better if authors mentions the parameters/factors to consider for survey, also mention the year wise segregation in a table . 

4. References cited in the paper are not the proper sequence order. These all should be order starting from [1], [2]…so on.

5. Some abbreviations are used in the paper but provided with their full forms, it should be rectified

6. Discuss major contributions of the paper clearly.

7. Overall flow and presentation of the paper can be improved.

8. Typos and grammatical mistakes should be removed.

9. Some of the figures are having low resolution, improve their quality. 

Author Response

Response to Reviewer 4’s Comments

Comment 1: In abstract " The experimental results show that the proposed conference recommendation method provides convenient and fast participation services as well as conference organization services. The developed big data platform can significantly improve the operation and participation efficiency of academic conferences, reduce the costs, and give full play to the role and value of academic conferences" there is no quantataive data here to prove that our model is better is how much percentage and how much does this can reduce cost?

Response 1: This paper mainly constructs a novel meeting recommendation platform based on user interest preference. At present, most of the existing service platforms only provide part of the services from the preparation to the end of the academic conference. For example, some platforms only provide solicitation services, and some platforms only provide conference publishing services. In addition, most of the organizers need to create their own conference websites for participants to register, and participants need to register on the corresponding websites of different conferences. As a result, the conference organizers and participants need to use multiple platforms to complete the conference process, which is very troublesome.

This paper designs and develops a comprehensive service platform for international academic conferences based on big data analysis. It can meet the requirements of conference organizers and participants for academic conferences in the Internet era and improve the quality and efficiency of academic conferences. Big data analysis technology can provide researchers with personalized meeting suggestions based on their research interests and meeting attendance records. It enables researchers to quickly find the meeting they are interested in and obtain the meeting information conveniently and quickly, so that researchers can manage and use their time more effectively in the fast-paced life.

Comment 2: In section 1, Different types of algorithms and techniques used for same issye is missing , add some more details in introduction part’?

Response 2: Thank you for your valuable comments and we are sorry for the unclear statement. We have revised this part in the manuscript, and the text of the revised manuscript is as follows:

There are already some platforms that provide services for academic conferences, For example, there are "AllConferences.Com", "World Conference Calendar", "AAAI Conferences", "EasyChair", "WikiCFP", "Cvent", etc. The conference organizer can quickly publish the solicitation information and participation information through this platform, and promote it, so that more scholars can see the information faster; Attendees can quickly access information about the conference they want to attend and register for the conference online.

However, most of the existing conference service platforms only provide part of the services mentioned above, rather than providing a "one-stop" service throughout the whole process of academic conferences, and some platforms are not strong across devices and international. For example, The WikiCFP platform mainly provides conference solicitation services, but does not support submission and review; The EasyChair platform mainly provides conference solicitation, online submission and review services, but does not support conference registration and registration services. In this way, the conference organizers release the conference, solicit and review manuscripts, and share conference materials through the cooperation of multiple platforms to complete, and most of the current practice of the conference organizers is to spend the cost of creating a dedicated site to collect information and charges for participants. Moreover, the operations of participants' conference information acquisition, submission, conference registration and registration also need to be completed on multiple platforms, leading to the cumbersome process of the conference organizer and the conference participant.

Comment 3: no contribution points can be seen in the mansucirpt ?.

Response 3: Academic conference can effectively promote academic exchanges among scholars, is a driving force for the development and progress of scientific research, and is of great benefit to the country, society and individual scholars. However, with the increase of the number of academic conferences, the quality of conferences and the efficiency of conference attendance have declined. In today's fast-paced life, high-quality and efficient academic conferences have become the first choice of scholars. Therefore, how to enable the conference organizers to hold academic conferences efficiently and enable the participating scholars around the world to quickly obtain relevant information about the conference and choose to attend the conference is the main research content of this paper.

This paper designs and develops the whole process of the big data platform for international academic conferences, from demand investigation and analysis to system testing and maintenance, and realizes an international-oriented academic conference service platform, which provides a series of convenient and quick services for conference organizers and participants, greatly improving the efficiency of academic conference and conference attendance and reducing costs. Give full play to the role and value of academic conferences. Big data analysis technology can provide researchers with personalized meeting suggestions based on their research interests and meeting attendance records. It enables researchers to quickly find the meeting they are interested in and obtain the meeting information conveniently and quickly, so that researchers can manage and use their time more effectively in the fast-paced life.

We have revised this part in the manuscript, and the text of the revised manuscript is as follows:

  • It provides a complete set of services related to academic conferences, such as conference release, conference classification and search, conference registration and payment, conference affairs management, etc., enabling the organizer to realize a "one-stop" for conference hosting and scholar attendance, reducing the cost of academic conference hosting and improving the efficiency of conference hosting and attendance.
  • Users can follow the meeting or meeting series that they are interested in. When users follow or register the meeting, the system will remind them of the relevant date of the corresponding meeting, such as the deadline for meeting solicitation, the date of meeting holding, etc. There are two reminding methods: system message reminder and email reminder, which can make scholars' time arrangement more reasonable and orderly. Effectively reduce the burden of scholars in the busy life.
  • A comprehensive evaluation of the conference and its series was conducted by using indicators such as the number of followers, page views, number of participants, rating of participants, and activity of the conference, so as to provide references for scholars to choose participants. The system can recommend personalized conferences to users according to their research interests, attendance records and meeting attention records, so that scholars can quickly find the academic conferences they are interested in. The system provides optional conference promotion services for the conference organizers, which makes the conference easier to attract users' attention, and promotes the promotion and influence of the conference.
  • The system uses lightweight development framework to achieve, quick response, simple and orderly interface, smooth transition effect, with excellent user experience. Responsive interface design can adapt to various resolutions of the device, convenient for users to access the system anytime and anywhere.

Comment 4: Section 2 presents some of the important literature work related to , it would be better if authors mentions the parameters/factors to consider for survey, also mention the year wise segregation in a table .

Response 4: Thank you for your valuable comments and we are sorry for the unclear statement. We have revised this part in the manuscript.

Ref.

Year

Viewpoint

[28]

2018

Allows data owners to outsource their data to remote storage resources provided by various storage providers.

[29]

2019

A data-driven approach is now a must-have tool for many IoT based services and applications.

[30]

2020

Decentralized storage powered by blockchain is becoming a new trend.

[31]

2020

The existence of wrong data will lead to the deterioration of the quality of big data.

[32]

2021

Based on research related to privacy issues and the relationship between data innovation through new services and applications in the Internet of Things environment.

[33]

2022

A data-driven approach is now becoming important to the business and the Internet of Things data market has emerged.

Comment 5: References cited in the paper are not the proper sequence order. These all should be order starting from [1], [2]…so on. 

Response 5: Thank you for your valuable comment. We have identified and arranged the order of references.

Comment 6: Some abbreviations are used in the paper but provided with their full forms, it should be rectified

Response 6: Thank you for your valuable comments. We have corrected all abbreviations for their full form.

Comment 7: Discuss major contributions of the paper clearly.

Response 7: Thank you for your valuable comments and we are sorry for the unclear statement. Your comments will be answered in the following aspects:

  • What is the motivation?

In recent years, the number of conferences held at home and abroad is increasing, and cross-international conferences are more and more favored by researchers. However, with the increase of the number of academic conferences, there appear some academic conferences with low quality and low cost performance. For example, some conferences have unclear themes, and the level of conference papers and reports is not high; The participation experience of scientific researchers is poor; Participants attend the conference in a short time and cannot fully communicate with other scholars at the conference site; The information sharing after the meeting is not convenient and timely; The quality and efficiency of some conference organizers are not high enough, which wastes a lot of conference resources, leading to lower cost performance of academic conferences. Therefore, how to improve the quality of academic conferences, reduce the cost of the organizers, simplify the process of participants, give full play to the role of academic exchanges, and maximize the value of academic conferences has become a problem to be solved.

  • What are the current challenges?

There are already some platforms that provide services for academic conferences. For example, there are "AllConferences.Com", "World Conference Calendar", "AAAI Conferences", "EasyChair", "WikiCFP", "Cvent", etc. The conference organizer can quickly publish the solicitation information and participation information through this platform, and promote it, so that more scholars can see the information faster; Attendees can quickly access information about the conference they want to attend and register for the conference online.

However, most of the existing conference service platforms only provide part of the services mentioned above, rather than providing a "one-stop" service throughout the whole process of academic conferences, and some platforms are not strong across devices and international. For example, The WikiCFP platform mainly provides conference solicitation services, but does not support submission and review; The EasyChair platform mainly provides conference solicitation, online submission and review services, but does not support conference registration and registration services. In this way, the conference organizers release the conference, solicit and review manuscripts, and share conference materials through the cooperation of multiple platforms to complete, and most of the current practice of the conference organizers is to spend the cost of creating a dedicated site to collect information and charges for participants. Moreover, the operations of participants' conference information acquisition, submission, conference registration and registration also need to be completed on multiple platforms, leading to the cumbersome process of the conference organizer and the conference participant.

3)  What is the contribution of this paper?

  • It provides a complete set of services related to academic conferences, such as conference release, conference classification and search, conference registration and payment, conference affairs management, etc., enabling the organizer to realize a "one-stop" for conference hosting and scholar attendance, reducing the cost of academic conference hosting and improving the efficiency of conference hosting and attendance.
  • Users can follow the meeting or meeting series that they are interested in. When users follow or register the meeting, the system will remind them of the relevant date of the corresponding meeting, such as the deadline for meeting solicitation, the date of meeting holding, etc. There are two reminding methods: system message reminder and email reminder, which can make scholars' time arrangement more reasonable and orderly. Effectively reduce the burden of scholars in the busy life.
  • A comprehensive evaluation of the conference and its series was conducted by using indicators such as the number of followers, page views, number of participants, rating of participants, and activity of the conference, so as to provide references for scholars to choose participants. The system can recommend personalized conferences to users according to their research interests, attendance records and meeting attention records, so that scholars can quickly find the academic conferences they are interested in. The system provides optional conference promotion services for the conference organizers, which makes the conference easier to attract users' attention, and promotes the promotion and influence of the conference.
  • The system uses lightweight development framework to achieve, quick response, simple and orderly interface, smooth transition effect, with excellent user experience. Responsive interface design can adapt to various resolutions of the device, convenient for users to access the system anytime and anywhere.

Comment 8: Overall flow and presentation of the paper can be improved.

Response 8: Thank you for your valuable comments and we are sorry for the unclear statement. We have revised section 1 in the manuscript, and the text of the revised manuscript is as follows:

1. Introduction

Big Data Analytics [1] is a term coined by researchers to describe the use of advanced analytical techniques that process, store, and collect very large, diverse big data sets for future examination. Data is being generated at an alarming rate, and the rapid progress of the Internet, Internet of Things (IoT), and other technologies [2] is the main reason behind this continued growth. The generated data reflects the environment in which it was generated, so we can use the data obtained from the system to figure out the internal work information of the system. In addition, the growth in data value makes big data a high-value target [3].

In the past 20 years, various applications have been developed and the data generated has grown exponentially, which has led to the advent of the era of big data. And big data analytics technology has been applied to many scenarios, such as topic analysis of online public opinion in social networks [4], traffic guidance [5], and personalized recommendation [6]. Academic Big Data is a collection of a large amount of academic information, technical data and partnership data, which have attracted increasing attention from industry and academia. The widespread adoption of the social computing paradigm has made it easier for researchers to join collaborative research activities and to share academic data more widely than ever before in highly intertwined academic networks [7].

1.1. Motivation

In recent years, the number of conferences held at home and abroad is increasing, and cross-international conferences are more and more favored by researchers. However, with the increase of the number of academic conferences, there appear some academic conferences with low quality and low cost performance. For example, some conferences have unclear themes, and the level of conference papers and reports is not high; The participation experience of scientific researchers is poor; Participants attend the conference in a short time and cannot fully communicate with other scholars at the conference site; The information sharing after the meeting is not convenient and timely; The quality and efficiency of some conference organizers are not high enough, which wastes a lot of conference resources, leading to lower cost performance of academic conferences. Therefore, how to improve the quality of academic conferences, reduce the cost of the organizers, simplify the process of participants, give full play to the role of academic exchanges, and maximize the value of academic conferences has become a problem to be solved.

1.2. Research Challenges

There are already some platforms that provide services for academic conferences. For example, there are "AllConferences.Com", "World Conference Calendar", "AAAI Conferences", "EasyChair", "WikiCFP", "Cvent", etc. The conference organizer can quickly publish the solicitation information and participation information through this platform, and promote it, so that more scholars can see the information faster; Attendees can quickly access information about the conference they want to attend and register for the conference online.

However, most of the existing conference service platforms only provide part of the services mentioned above, rather than providing a "one-stop" service throughout the whole process of academic conferences, and some platforms are not strong across devices and international. For example, The WikiCFP platform mainly provides conference solicitation services, but does not support submission and review; The EasyChair platform mainly provides conference solicitation, online submission and review services, but does not support conference registration and registration services. In this way, the conference organizers release the conference, solicit and review manuscripts, and share conference materials through the cooperation of multiple platforms to complete, and most of the current practice of the conference organizers is to spend the cost of creating a dedicated site to collect information and charges for participants. Moreover, the operations of participants' conference information acquisition, submission, conference registration and registration also need to be completed on multiple platforms, leading to the cumbersome process of the conference organizer and the conference participant.

1.3. Contributions

  • It provides a complete set of services related to academic conferences, such as conference release, conference classification and search, conference registration and payment, conference affairs management, etc., enabling the organizer to realize a "one-stop" for conference hosting and scholar attendance, reducing the cost of academic conference hosting and improving the efficiency of conference hosting and attendance.
  • Users can follow the meeting or meeting series that they are interested in. When users follow or register the meeting, the system will remind them of the relevant date of the corresponding meeting, such as the deadline for meeting solicitation, the date of meeting holding, etc. There are two reminding methods: system message reminder and email reminder, which can make scholars' time arrangement more reasonable and orderly. Effectively reduce the burden of scholars in the busy life.
  • A comprehensive evaluation of the conference and its series was conducted by using indicators such as the number of followers, page views, number of participants, rating of participants, and activity of the conference, so as to provide references for scholars to choose participants. The system can recommend personalized conferences to users according to their research interests, attendance records and meeting attention records, so that scholars can quickly find the academic conferences they are interested in. The system provides optional conference promotion services for the conference organizers, which makes the conference easier to attract users' attention, and promotes the promotion and influence of the conference.
  • The system uses lightweight development framework to achieve, quick response, simple and orderly interface, smooth transition effect, with excellent user experience. Responsive interface design can adapt to various resolutions of the device, convenient for users to access the system anytime and anywhere.

Comment 9:  Typos and grammatical mistakes should be removed.

Response 9: Thank you for your valuable comments and we have invited a native speaker to help us polish the presentation of this paper.

Comment 10: Some of the figures are having low resolution, improve their quality.

Response 10: Thank you for your valuable comments. We have deleted Figures 5, 6, and 11 which have less information and low resolution.